# Collagen–Platelet-Rich Plasma Mixed Hydrogels as a pBMP2 Delivery System for Bone Defect Regeneration

**DOI:** 10.3390/biomedicines12112461

**Published:** 2024-10-26

**Authors:** Anastasiia Yurevna Meglei, Irina Alekseevna Nedorubova, Viktoriia Pavlovna Basina, Viktoria Olegovna Chernomyrdina, Andrey Anatolevich Nedorubov, Valeriya Sergeevna Kuznetsova, Andrey Vyacheslavovich Vasilyev, Sergey Ivanovich Kutsev, Dmitry Vadimovich Goldshtein, Tatiana Borisovna Bukharova

**Affiliations:** 1Research Centre for Medical Genetics, 115478 Moscow, Russia; nedorubova.ia@gmail.com (I.A.N.); vika.basina12@gmail.com (V.P.B.); victoria-mok@yandex.ru (V.O.C.); tilia7@yandex.ru (V.S.K.); vav-stom@yandex.ru (A.V.V.); kutsev@mail.ru (S.I.K.); dvgoldshtein@gmail.com (D.V.G.); bukharova-rmt@yandex.ru (T.B.B.); 2Central Research Institute of Dental and Maxillofacial Surgery, 119021 Moscow, Russia; 3Institute of Translational Medicine and Biotechnology, Sechenov University, 119991 Moscow, Russia; nedorubov.ras@gmail.com

**Keywords:** gene-activated matrices, BMP2, non-viral delivery, collagen I, platelet-rich plasma, critical size bone defect, bone regeneration

## Abstract

**Background/Objectives:** The replenishment of bone deficiency remains a challenging task in clinical practice. The use of gene-activated matrices (GAMs) impregnated with genetic constructs may be an innovative approach to solving this problem. The aim of this work is to develop collagen-based matrices with the addition of platelet-rich plasma, carrying polyplexes with the *BMP2* gene, to study their biocompatibility and osteogenic potential in vitro and in vivo. **Methods:** The cytocompatibility of the materials during incubation with adipose-derived stem cells (ADSCs) was studied using the MTT test and fluorescent microscopy. Biocompatibility was assessed during intramuscular implantation, followed by histological analysis. Osteogenic differentiation was determined by the expressions of *Alpl* and *Bglap* using real-time PCR and extracellular matrix (ECM) mineralization by alizarin red staining. The efficiency of bone regeneration was studied using micro-CT and analysis of histological sections stained according to Masson. **Results:** After the incubation of ADSCs with GAS, significant increases in the expressions of the *Alpl* and *Bglap* genes by 3 ± 0.1 and 9.9 ± 0.6 times, relative to the control, as well as mineralization of the ECM, were observed. The volume of newly formed bone was 37.2 ± 6.2% after implantation of GAS, 20.9 ± 1.2%—non-activated Col/PRP, and 2.6 ± 1.5% in an empty defect. **Conclusions:** The use of Col/PRP-based matrices is an effective method for delivering of the osteoinductor gene to the site of bone tissue damage. The highest degree of healing was observed after the implantation of Col/PRP-TF/pBMP2 into the critical size defect compared to the other groups.

## 1. Introduction

The restoration of bone volume deficiency caused by trauma, tumor resection, and the treatment of congenital anomalies is a common problem in surgery, orthopedics, and dentistry. Autotransplantation remains the gold standard for the treatment of bone defects. Autogenous bone graft has osteogenic and osteoinductive effects due to the contents of mesenchymal stem cells (MSCs) and growth factors, which functions as an osteoconductive scaffold for new bone growth [1]. However, obtaining autografts is associated with some limitations and complications, which require the search for alternative approaches.

One of them is the development of scaffolds from various natural and synthetic biomaterials [2]. Scaffolds can mimic the porous structure of cancellous bone tissue [3,4,5] or imitate the extracellular matrix of maturing bone [6,7]. Scaffolds are used as temporary implants; therefore, they must be biocompatible and biodegradable, as well as promote osteogenic processes in the defect area.

Type I collagen (Col) is a natural polymer that is a major component of the bone matrix [8]. Collagen-based materials are widely used in bone tissue engineering [9]. Single-component extracellular matrix analogues are not able to mimic the complex osteogenic microenvironment [10], which determines the use of additional components that can enhance their regenerative properties.

One such component may be platelet-rich plasma (PRP). Over the past decades, interest in PRP as an effective material in regenerative medicine has increased significantly [11]. PRP has regenerative potential due to its content of growth factors that promote the angiogenesis, proliferation, and migration of stem cells to the site of injury. Moreover, PRP is an autogenous product that does not cause any adverse reactions [12].

To increase the efficiency of osteoprogenitor cell differentiation at the site of injury, scaffolds are used in combination with osteoinductive factors. The most widely studied and effective osteoinducer is bone morphogenetic protein 2 (BMP-2), from the transforming growth factor (TGF-β) superfamily [13]. BMP-2 plays a significant role in = osteoinduction, essential for new bone formation. It acts as a chemokine for osteoprogenitor cells and promotes their differentiation [14]. The US Food and Drug Administration has approved the use of recombinant human BMP-2 (rhBMP-2) in clinical practice. However, the low stability of the protein molecules requires the administration of supraphysiological doses to achieve a therapeutic effect. The high doses and bolus type of growth factor release can lead to adverse reactions [15]. Gene therapy based on the delivery of osteoinducer genes as a part of vector systems provides protein synthesis in resident cells in physiological doses and appropriate time settings [16]. Viral and non-viral vectors as parts of matrices are used to deliver target genes to cells. Non-viral vectors are considered the safest, since the use of such systems reduces the risk of insertional mutagenesis and the host immune response [17]. Plasmid vectors, delivered into cells using various transfection agents, are widely used in genetic engineering [18].

Thus, the aim of this study is to develop gene-activated matrices based on collagen with the addition of PRP, carrying polyplexes with the *BMP2* gene, and to study their biocompatibility and osteogenic potential in vitro and in vivo.

## 2. Materials and Methods

### 2.1. Materials

To obtain Col/PRP matrices, hydrogel based on type I collagen derived from the dermis of cattle (Col; Intens, Biopharmaholding, Moscow, Russia) was mixed with PRP, and then thrombin (100 NIH; PZ Cormay, Warsaw, Poland) in a 10% calcium chloride solution (NPO Microgen, Moscow, Russia) was added drop by drop for polymerization. PRP was obtained from rat’s blood using a previously described method [7]. The platelet content in PRP was 4.3 ± 0.6 times higher than that in whole blood. The ratio of Col and PRP was 1:2 by volume. The structure of the materials was assessed with histological preparations and staining them with hematoxylin and eosin according to the standard method described by the manufacturer of the kit (Biovitrum, St. Petersburg, Russia).

Plasmids carrying the target *BMP2* gene (pBMP2; pTagRFP-N-BMP2, Evrogen, Moscow, Russia) were used to obtain gene-activated matrices. The plasmid DNA was constructed based on the pTagRFP-N vector, with the *RFP* gene replaced with the target *BMP2* gene. Plasmids were grown in *Escherichia coli* cells in LB broth medium (Sigma-Aldrich, St. Louis, MO, USA) with 50 μg/mL kanamycin (GRiSP, Porto, Portugal) and were isolated using a Plasmid Midiprep kit (Evrogen, Moscow, Russia) according to the manufacturer’s protocol. To form polyplexes, 40 μL/mL TurboFect (TF, Thermo Fisher Scientific, Waltham, MA, USA) and 20 μg/mL plasmids were incubated for 20 min at 37 °C. The resulting polyplexes were mixed with 50 μL of type I collagen hydrogel and incubated for 1 h, then 120 μL of PRP and 30 μL of thrombin solution were added for gelling.

### 2.2. Cell Cultures

The culture of mesenchymal stromal cells was isolated from rat adipose tissue (ADSCs) using a previously developed method [19]. The cells were cultured in a growth medium, DMEM/F12 (PanEko, Moscow, Russia), containing 10% fetal bovine serum (FBS, PAA Laboratories, Etobicoke, ON, Canada), 0.584 mg/mL L-glutamine (PanEko, Moscow, Russia), 5000 U/mL penicillin (PanEko, Moscow, Russia), and 5000 μg/mL streptomycin (PanEko, Moscow, Russia) at 37 °C and 5% CO_2_. The growth medium was changed every 3 days.

ADSCs were incubated in the presence of matrices in 24-well plates with a Transwell system (Corning, NY, USA). Cell cultures were seeded at the bottom of the wells, and the test material was placed in filters after 24 h.

To assess the osteogenic properties of the gene-activated matrices, ADSCs were cultured in osteogenic medium containing DMEM (PanEko, Moscow, Russia), 10% FBS, 0.584 mg/mL L-glutamine, 0.05 mg/mL L-ascorbic acid (Sigma, St. Louis, MO, USA), 2.16 mg/mL β-glycerophosphate (Sigma), 5000 U/mL penicillin, and 5000 μg/mL streptomycin at 37 °C and 5% CO_2_ for 14 days; half of the medium was changed every 3 days.

### 2.3. Assessment of Matrix Cytocompatibility

Cell viability was assessed by MTT test on days 1 and 7 after incubation. For this purpose, at the end of the experiment, 0.5 mg/mL 3-(4,5-dimethylthiazol-2-yl)-2,5-diphenyltetrazolium bromide (MTT; PanEco, Moscow, Russia) was added to the wells with ADSCs and incubated for 2 h at 37 °C. Then, formazan crystals were dissolved using DMSO (PanEco, Moscow, Russia) and mixed on a shaker for 10 min, and the optical density of the eluate was measured on an xMark plate spectrophotometer (Bio-Rad, Hercules, CA, USA) at a wavelength of 570 nm, subtracting the background value at 620 nm.

The cytotoxicity of the matrices was assessed on days 1 and 7 after incubation with ADSCs by fluorescence microscopy using a Lionheart FX automated microscope (Agilent BioTek, Santa-Clara, CA, USA). Cells without matrices were used as a control. For this study, cells pre-stained with РКН-26 (red fluorescent cell linker kit, Sigma, St. Louis, MO, USA) in accordance with the manufacturer’s instructions were seeded in the wells of a 24-well plate with the studied materials. At each stage, to identify dead cells, ADSCs were stained with fluorescent dye, DAPI (4,6-diamidino-2-phenylindole), at a concentration of 5 μg/mL for 10 min at room temperature, and living cells were stained with 0.5 μM Calcein AM (Biotium, Fremont, CA, USA) for 35 min at 37 °C.

### 2.4. Kinetics of Plasmid DNA Release

Col/PRP materials, as well as each component separately, were impregnated with plasmid DNA with the *BMP2* gene and incubated in a physiological solution (PanEko, Moscow, Russia), and the release of plasmid DNA was assessed for 21 days. The concentration of plasmid DNA was determined by spectrophotometry at a wavelength of 260 nm on a NanoDrop OneC spectrophotometer (Thermo Fisher Scientific, Waltham, MA, USA).

### 2.5. Real-Time PCR

Materials impregnated with TF/pBMP2 polyplexes were incubated with ADSCs in an osteogenic medium at 37 °C for 14 days; half of the medium was changed every 3 days. As a negative control, we used cells that had been cultured in the presence of non-activated Col/PRP materials. The analysis was performed at the end of the incubation period.

The expression of the *BMP2*, alkaline phosphatase (*Alpl*), and osteocalcin (*Bglap*) genes was assessed using the intercalating dye SYBR Green, with preliminary reverse-transcription reaction. Total RNA was isolated from the cells with an RNeasy Plus Mini Kit according to the manufacturer’s recommendations (Qiagen, Hilden, Germany). cDNA was synthesized using a RevertAid First-Strand cDNA Synthesis Kit with Reverse Transcriptase (Thermo Fisher Scientific, Waltham, MA, USA). The obtained values were normalized relative to the reference genes: *Gapdh* and *Actβ*. Primers (Evrogen, Moscow, Russia) for the studied genes are shown in Table 1.

### 2.6. ELISA Assay

To evaluate the production of BMP-2 protein secreted by the ADSCs, the medium was collected throughout the entire incubation period of the cells with the studied matrices and stored at −80 °C. Then, all fractions were combined; the protein was concentrated using Amicon centrifugal filters (Merck KGaA, Darmstadt, Germany) and examined with an enzyme immunoassay using a Quantikine Elisa kit (R&D Systems, Minneapolis, MN, USA), according to the manufacturer’s protocol. Measurements were performed on an xMark plate spectrophotometer (Bio-Rad, Hercules, CA, USA).

### 2.7. Alizarin Red Staining

For analysis of ECM mineralization, cells were fixed with cold 70% ethanol for 30 min at 4 °C, then stained with 2% aqueous alizarin red (Sigma-Aldrich, St. Louis, MO, USA) at pH 4.1 for 5 min. The culture was washed twice with distilled water to remove unbound dye, and images of the cells were obtained using light microscopy.

### 2.8. In Vivo Study

In vivo experiments were performed on male Wistar rats weighing 250–300 g. Each experimental group consisted of six animals. All in vivo experiments were approved by the local Bioethics Committee of Sechenov University (No. PRC-079 dated 6 April 2021) in accordance with the Guide for the Care and Use of Laboratory Animals published by the US National Institutes of Health (NIH publication No.85–23, revised 1996), the European Convention for the Protection of Vertebrate Animals used for Experimental and Other Scientific Purposes, and International Organization for Standardization [20]. Before matrix implantation, rats were anesthetized with an intramuscular injection of zoletil at a dose of 30 mg/kg (Virbac, Carros, France) and 5 mg/kg xylazine (InterchemieWerken “de Adelaar” BV, Venray, The Netherlands). At the end of the experiments, the animals were euthanized by CO_2_ inhalation. Biopsies of the implantation area were obtained and fixed in 10% neutral formalin solution (Labiko, St. Petersburg, Russia) for at least 24 h.

The biocompatibility of the materials in vivo was studied by the intramuscular implantation of the samples into the posterior surface of the thigh of the rats. The implantation area was shaved and disinfected, a longitudinal skin incision was made, and the matrices were placed at a depth of 1.5–2 cm between the muscle heads. The skin was sutured with Vicryl 5/0 (Ethicon, Raritan, NJ, USA). Histological analysis was performed 14 days after implantation.

The efficiency of bone regeneration in vivo was studied on a model of a critical-size defect of the parietal bones of rats 8 weeks after surgery. A 7 mm diameter cranial bone defect was obtained using a Creamer trephine (LZQ, Seoul, Republic of Korea) with regular irrigation with sterile saline. Matrices were implanted into the bone defect and the skin was sutured with 5/0 Vicryl. Two control groups were used in this experiment. In the first group, animals underwent craniotomy without subsequent implantation of matrices (empty). The second group underwent implantation of materials without a plasmid vector. The efficiency of osteogenesis was assessed using micro-CT and histological analysis.

### 2.9. Micro-CT

The biopsies were examined using a SkyScan 1276 high-resolution micro-CT scanner (Bruker, Kontich, Belgium) with an X-ray output of 60 kV. NRecon reconstruction software (version 1.6.10.2, Bruker, Belgium) was used to create 3D reconstructions from the scanned images. The resulting images were analyzed using Dragonfly software v2022.1 (Montreal, QC, Canada), and new bone volume (Nb.V.%) was measured.

### 2.10. Histological Assay

Formaldehyde-fixed samples were dehydrated in an alcohol-xylene gradient and embedded in paraffin. Bone biopsy fragments were preliminarily decalcified in Trilon-B (BioVitrum, St. Petersburg, Russia) for 3 weeks. Sections 5–10 μm thick were prepared, muscle samples were stained with hematoxylin and eosin, and bone biopsy samples were stained with Masson trichrome (BioVitrum, St. Petersburg, Russia) according to the manufacturer’s protocol. Histological preparations were examined using an Axio Observer D1 microscope with an AxioCam HRc Axioimager M.1 camera (Carl Zeiss, Oberkochen, Germany).

### 2.11. Statistical Analysis

Plotting of graphs and statistical analysis were performed in SigmaPlot v14.0 (Systat Software Inc., Palo Alto, CA, USA). All data are presented as mean ± SD. Groups were compared using Student’s t-test or Mann–Whitney U-test. For multiple comparisons of groups, the one-way ANOVA Holm–Sidak test was used. Differences were considered statistically significant at a level of statistical significance below 5% (*p* < 0.05). When plotting bar charts, values that differed from the control were marked with an asterisk “*” in accordance with the recommendations of the American Physiological Association (APA).

## 3. Results

### 3.1. Col/PRP Materials Formation

Collagen hydrogel is unable to retain its shape. Adding PRP to Col made it possible to obtain a denser and resilient material (Figure 1a). Hematoxylin and eosin staining allowed the evaluation of the structure of the materials (Figure 1b). We observed a uniform distribution of components in the structure of the matrices.

### 3.2. Biocompatibility of Matrices In Vitro

Fluorescence microscopy revealed that most cells were stained with Calcein AM, which confirmed their viability, and practically no dead cells stained with DAPI were detected (Figure 2a). The relative number of living cells in the presence of the matrices on day 7 was significantly higher than in the control, reaching 114.9 ± 5.7% in the Col/PRP group and 124.6 ± 11.3% in the Col/PRP-TF/pBMP2 group (Figure 2b). The increase in the number of living cells after incubation with matrices compared to the control was probably caused by the influence of growth factors included in the PRP [21]. No statistically significant differences were found between the groups with the materials. Thus, the biocompatibility of the studied matrices and the positive effect of the material components on the proliferation of ADSCs were demonstrated.

### 3.3. Biocompatibility of Materials In Vivo

The histological analysis on the 14th day after the intramuscular implantation of Col/PRP materials showed that giant cells of foreign bodies (GCs) were present at the border of the materials. During implantation, fragments of non-resorbed materials and the formation of granulation tissue with blood vessels were observed. There were no signs of pronounced acute exudative inflammation (Figure 3). Thus, the developed materials are biocompatible and biodegradable.

### 3.4. Transfection Efficiency of ADSCs with TF/pBMP2 Polyplexes Impregnated into Col/PRP Materials

The transfection capacity of TF/pBMP2-Col/PRP was assessed by the level of *BMP2* gene expression and by the production of BMP-2 protein in ADSCs after incubation with the matrices. Materials without plasmids (Col/PRP) were added to the control group. It was shown that the expression of the *BMP2* gene and the production of BMP-2 protein in the cells significantly increased relative to those of the control by 2737 ± 84.1 and 24 ± 0.32 times, respectively (Figure 4). 

### 3.5. Kinetics of pBMP2 Release from Matrices

It was shown that the release of plasmids from the Col/PRP-pBMP2 and PRP-pBMP2 matrices occurred comparably and gradually over 3 weeks (Figure 5). At the same time, plasmid DNA impregnated in the Col-pBMP2 matrices was completely released on the seventh day. Thus, the addition of PRP to collagen promotes a longer release of plasmid DNA.

### 3.6. Osteogenic Differentiation of ADSCs After Incubation with Col/PRP-TF/pBMP2 Materials

Cultivation of ADSCs in the presence of materials containing polyplexes with the *BMP2* gene was accompanied by a statistically significant increase in alkaline phosphatase and osteocalcin genes expression by 3 ± 0.09 and 9.9 ± 0.6 times, respectively, compared to the control (Figure 6a). The EMC mineralization of ADSCs was also revealed when incubating cells with GAS (Figure 6b). Thus, the cultivation of ADSCs with Col/PRP-TF/pBMP2 promoted the induction of osteogenic differentiation of cells.

### 3.7. Bone Regeneration In Vivo

Eight weeks after the matrices were implanted into the critical-size defect of the rat parietal bones, new bone was formed from the maternal bone, and fibrous connective tissue was observed in the defect area (Figure 7). The absence of a pronounced inflammatory reaction in all groups further demonstrated the excellent biocompatibility of the matrices. Masson staining of the sections showed that in the group with GAS, a significant area of the defect was filled with mature bone tissue compared to the empty defect and when using non-activated Col/PRP materials (Figure 7b). The volume of all newly formed bone (Nb.V.%) was 37.2 ± 6.2% after implantation of GAS, 20.9 ± 1.2% for non-activated Col/PRP materials, and 2.6 ± 1.5% in the empty defect (Figure 7c). Thus, the inclusion of plasmids with the *BMP2* gene into materials stimulates reparative processes in bone defect areas.

## 4. Discussion

In the last two decades, the development of gene-activated matrices, which are both carriers of vector systems and matrices for the formation of new tissues, has been active. To obtain matrices with high regenerative efficiency, it is necessary to rely on natural processes during bone tissue restoration. In our study, hydrogel materials based on type I collagen and PRP impregnated with the *BMP2* gene were developed. The fibrin clot within the material’s structure mimics a hematoma that forms at an early stage of bone defect healing. The formation of a hematoma allows for the initiation of an inflammatory cascade, accompanied by the recruitment of proinflammatory cells; the sequential secretion of inflammatory, chemotactic, and progenitor factors; and, as a result, the attraction of resident cells. The combination of these events leads to hematoma resorption, bone callus formation, and osteoblastic cell differentiation [1].

The biomimetic structure of hydrogel materials promotes cell adhesion and proliferation [22]. Studies have shown that Col/PRP materials are biocompatible, support ADSC proliferation, and are resorbable, as demonstrated in an in vivo experimental model. Resorption of the implanted materials is important for the formation of new bone tissue, as well as for the release of the bioactive molecules included in the matrices [23].

In our study, a plasmid vector encoding the *BMP2* gene was used to provide osteoinduction. Vector systems are more effective than recombinant proteins in the long term [16]. Col/PRP materials impregnated with *BMP2* polyplexes stimulated the osteoblastic differentiation of ADSCs (Figure 6). After two weeks of the cultivation of cells with GAS, significant increases in the expressions of osteogenic markers and ECM mineralization were observed. Scaffolds based on natural biopolymers in combination with vector systems carrying *BMP2* demonstrated excellent osteogenic potential when studied on the MC3T3-E1 cell line [24]. At the same time, in our in vitro model using Col/PRP-TF/pBMP2 matrices, more pronounced signs of mineralization were observed, which was reflected in a higher level of expression of osteocalcin, one of the markers of late osteoblastic differentiation.

The composition of a matrix has a significant effect on the rate of bone regeneration in the critical-size defect area. A number of studies demonstrate the regenerative potential of PRP implanted into bone defects [21], and collagen itself is able to stimulate the osteogenic differentiation of MSCs [8]. It has been previously shown that collagen-based scaffolds with nanohydroxyapatite with dual delivery of *BMP2* and *VEGF* genes contributed to defect healing by 10% and by 5% in the case of using non-activated materials 4 weeks after implantation [25]. In our study, 8 weeks after implantation, non-activated Col/PRP materials provided a 20% decrease in the defect area. However, the most pronounced osteogenic effect was observed when plasmids with the *BMP2* gene were added to the materials, which contributed to the healing of the bone defect by 37% (Figure 7). The use of PRP in the materials probably helps to provide the defect area with growth factors at the earliest stage, which stimulates regenerative processes. Also, the addition of PRP to the collagen scaffold makes the hydrogel denser; allows it to maintain the scaffold shape for several weeks [7]; and, therefore, promotes a longer release of plasmid DNA at the end of the acute inflammatory stage, vessel growth, and migration of osteoprogenitor cells. Vector systems with the *BMP2* gene in combination with biopolymer-based scaffolds are highly effective when applied to metal implants, as demonstrated in a rat mandibular defect [26]. The authors showed that adding vector systems with *BMP2* to a titanium scaffold with PDLLA contributed to the uniform filling of defects with newly formed bone, compared with a non-activated scaffold, which is comparable to our results. Probably, the gradual release of the plasmid vector with the *BMP2* gene over a long period can promote prolonged transfection of resident cells and increase the efficiency of reparative osteogenesis due to paracrine stimulation, including endogenous BMP-2. However, the undoubted advantage of the developed GAS Col/PRP-TF/pBMP2 is its biomimetic structure, as well as the availability of the components used.

## 5. Conclusions

The obtained results illustrate the efficiency of the developed gene-activated Col/PRP-pBMP2/TF matrices. The controlled and localized release of the plasmid vector is one of the factors influencing the efficiency of reparative processes in critical defect areas. The high biocompatibility and osteogenic properties of the developed matrices may allow their use as working drugs for the treatment of bone defects, capable of replacing the conventional osteoplastic materials currently used.

## Figures and Tables

**Figure 1 biomedicines-12-02461-f001:**
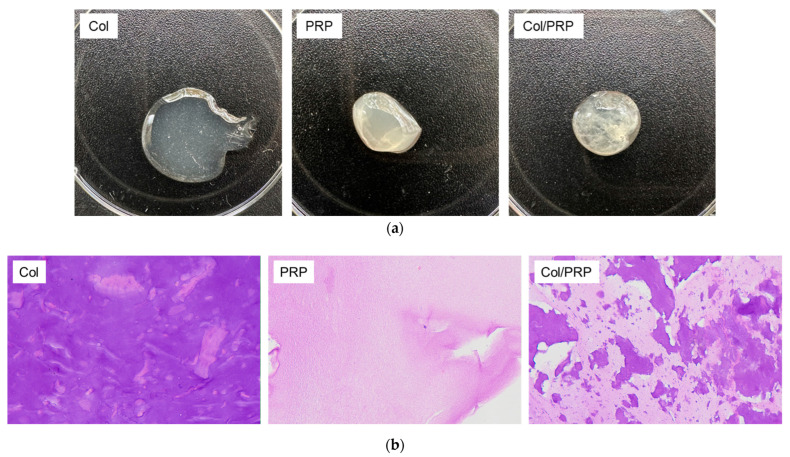
Col/PRP materials: (**a**) appearance of the materials; and (**b**) material’s structure, hematoxylin–eosin staining, light microscopy. Magnification 40×.

**Figure 2 biomedicines-12-02461-f002:**
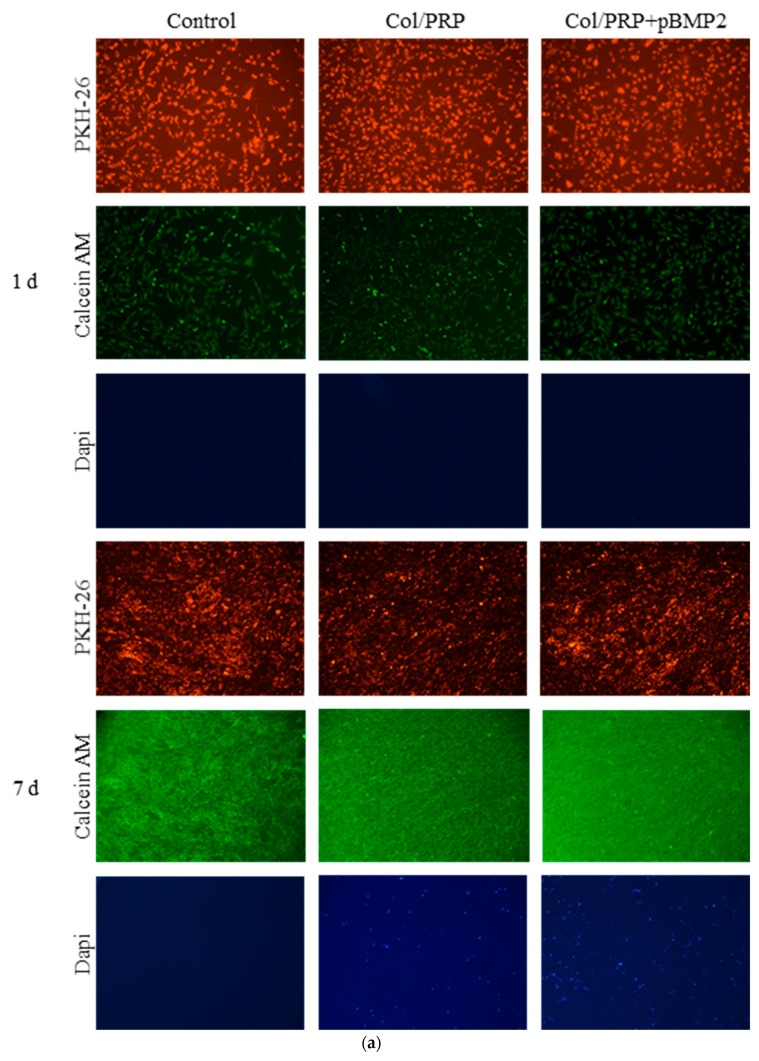
Evaluation of biocompatibility of Col/PRP-TF/pBMP2 matrices after incubation with ADSCs: (**a**) ADSCs stained with RKH-26 (red), Calcein AM (live cells; green), and DAPI (dead cells; blue), fluorescence microscopy. Magnification 10×; (**b**) relative cell viability, MTT test. * *p* < 0.05 compared to control.

**Figure 3 biomedicines-12-02461-f003:**
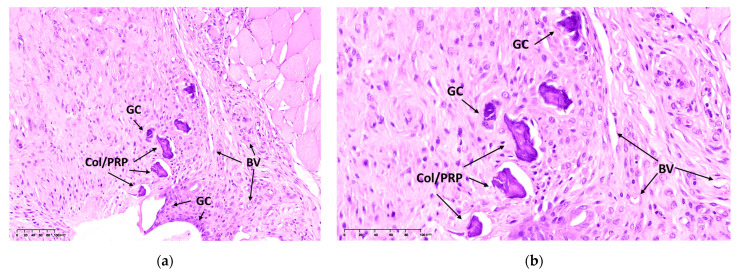
Evaluation of the biocompatibility of Col/PRP materials 14 days after intramuscular implantation in rats. Hematoxylin–eosin staining, light microscopy. Magnification: (**a**) 20×; (**b**) 40×. BV—blood vessel, GC—foreign body giant cell.

**Figure 4 biomedicines-12-02461-f004:**
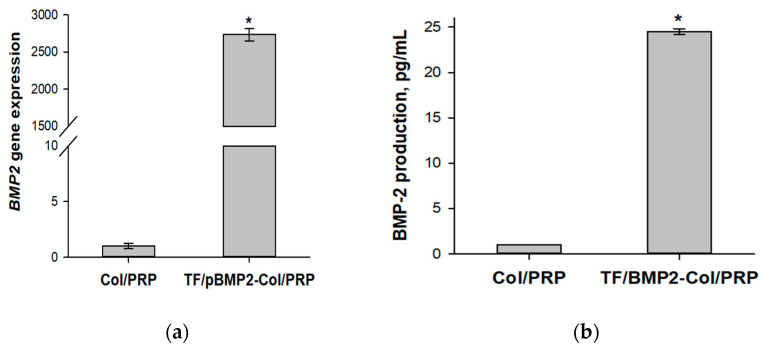
Analysis of the efficiency of ADSCs’ transfection with polyplexes containing the *BMP2* gene impregnated into Col/PRP materials: (**a**) relative expression of the *BMP2* gene, real-time PCR; (**b**) production of BMP-2 protein, ELISA. * *p* < 0.05 compared to Col/PRP.

**Figure 5 biomedicines-12-02461-f005:**
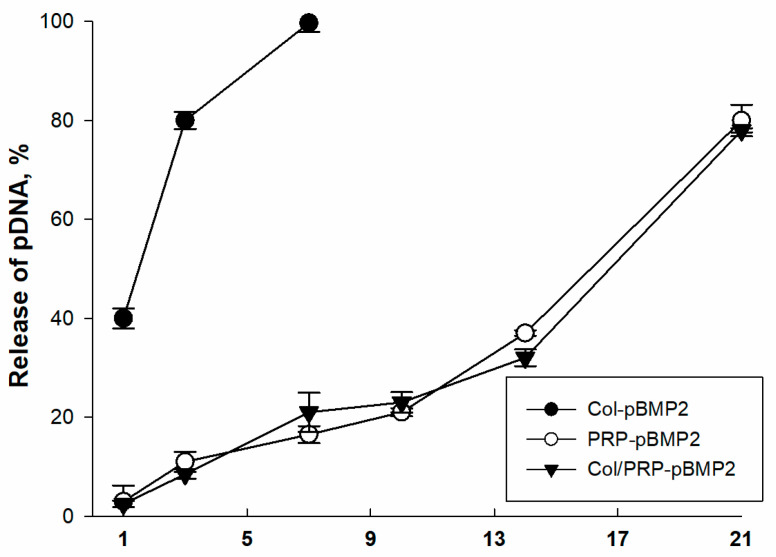
Kinetics of plasmid DNA release from Col/PRP materials and its components. Spectrophotometry.

**Figure 6 biomedicines-12-02461-f006:**
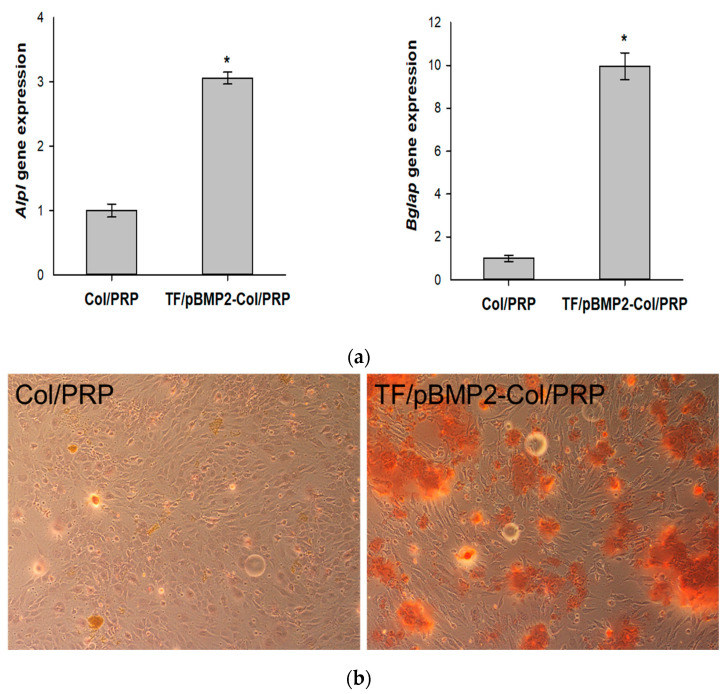
Analysis of osteogenic differentiation of ADSCs 14 days after incubation with Col/PRP-TF/pBMP2 matrices: (**a**) relative expression of *Alpl* and *Bglap* genes, real-time PCR. * *p* < 0.05 compared with Col/PRP; (**b**) ECM mineralization of ADSCs. Alizarin red staining, light microscopy. Magnification 10×.

**Figure 7 biomedicines-12-02461-f007:**
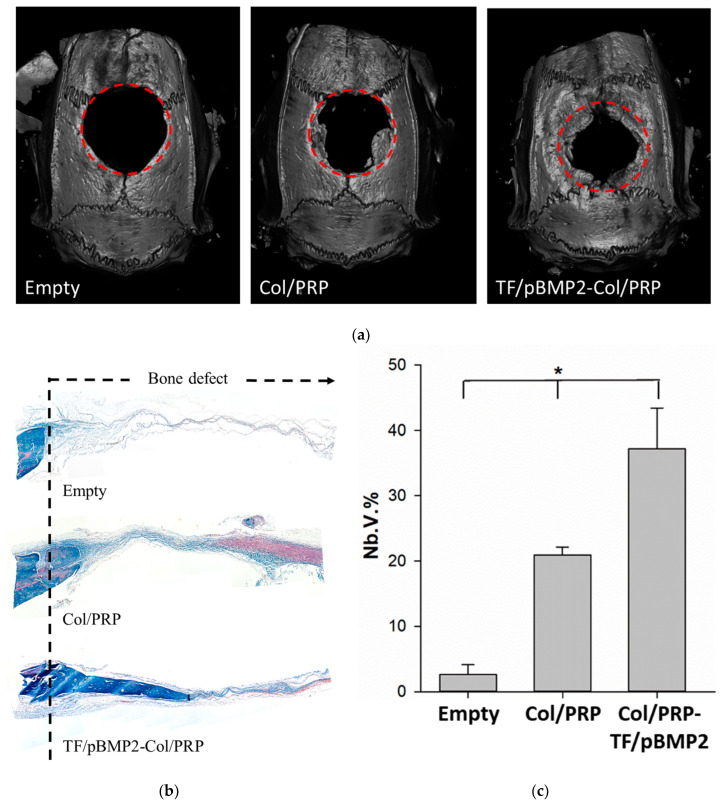
Regeneration of critical-size parietal bone defect in rats 56 days after implantation of Col/PRP-TF/pBMP2; (**a**) images obtained using a micro-CT scanner. The red dotted line marks the border of the bone defect; (**b**) histological preparations stained with Masson with aniline blue, light microscopy. Magnification 5×; (**c**) volume of newly formed bone (Nb.V.%), assessed using micro-CT. * *p* < 0.05 compared to empty defect.

**Table 1 biomedicines-12-02461-t001:** Sequence of primers used in RT-PCR.

Gene	Primer Sequence *, 5′→3′
*Actβ*	For: GAGATTACTGCCCTGGCTCC Rev: GCTCAGTAACAGTCCGCCTA
*BMP2*	For: ACTACCAGAAACGAGTGGGAA Rev: GCATCTGTTCTCGGAAAACCT
*Bglap*	For: CCTAGCAGACACCATGAGGAC Rev: CAGGTCAGAGAGGCAGAATGC
*Gapdh*	For: GCGAGATCCCGCTAACATCA Rev: CCCTTCCACGATGCCAAAGT

* Note: For—forward primer; Rev—reverse primer.

## Data Availability

Data are contained within the article.

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
