# Peer review of "Collagen–Platelet-Rich Plasma Mixed Hydrogels as a pBMP2 Delivery System for Bone Defect Regeneration"

_biomedicines, 2024, doi:10.3390/biomedicines12112461_

Round 1
Reviewer 1 Report
Comments and Suggestions for Authors
1. In Materials and Methods line no 79 Plasmids carrying the target BMP2 gene (pBMP2; pTagRFP-N-BMP2), platelet-rich plasma (PRP) is only mentioned above but, in the composition, it is RFP. What is RFP?
2. The CO2 subscript is not in correct manner correct it
3. The given date is ok but add the material photos
4. Give the data about the physical properties of the hydrogel in addition to its biological properties.
5. Assays tests are important but give other related datas to have the characteristic behaviour of the materials.
6. Add More data about the bone defect regeneration like growth factor and its physical nature shifting’s
Author Response
Response to Reviewer
We would like to thank the reviewer for the useful comments and suggestions on our manuscript biomedicines-3242502, entitled “Collagen-PRP Mixed Hydrogels as a pBMP2 delivery system for bone defect regeneration”. We have checked and revised the manuscript carefully according to the comments and suggestions, and the detailed responses have been listed below with all the changes and improvements included in the revised manuscript.
Comments 1. In Materials and Methods line no 79 Plasmids carrying the target BMP2 gene (pBMP2; pTagRFP-N-BMP2), platelet-rich plasma (PRP) is only mentioned above but, in the composition, it is RFP. What is RFP?
Response 1: Plasmids with the BMP2 gene (pTagRFP-N-BMP2) were constructed based on the pTagRFP-N vector at Evrogen. To achieve this, the BMP2 gene was inserted instead of the Red Fluorescent Protein (RFP) gene. We added explanations in the text of the article: “The plasmid DNA was constructed based on the pTagRFP-N vector, with the RFP gene replaced by the target BMP2 gene”.
Comments 2. The CO2 subscript is not in correct manner correct it
Response 2: Thank you for pointing this out. We have corrected the CO2 subscript.
Comments 3. The given date is ok but add the material photos
Response 3: We have added Figure 1, which shows the appearance of the material and its structure and explanations to it.
Comments 4. Give the data about the physical properties of the hydrogel in addition to its biological properties.
Response 4: In this study, we did not set the goal of obtaining a material similar in physical and mechanical parameters to bone tissue. We developed a carrier matrix for delivering the vector systems, which should support the new bone formation. We added an image of the material (Figure 1) and showed that by mixing the components we got a more formed and elastic material.
Comments 5. Assays tests are important but give other related datas to have the characteristic behaviour of the materials.
Response 5: We have added information on the structure of the studied materials, obtained by histological staining (Figure 1).
Comments 6. Add More data about the bone defect regeneration like growth factor and its physical nature shifting’s.
Response 6: The introduction has been updated to include more data on the role of BMP-2 in bone defect regeneration: “To increase the efficiency of osteoprogenitor cell differentiation at the site of injury, scaffolds are used in combination with osteoinductive factors. The most widely studied and effective osteoinducer is bone morphogenetic protein 2 (BMP-2) from the transforming growth factor (TGF-β) superfamily [13]. BMP-2 plays a significant role in the osteoinduction, essential for new bone formation. They act as chemokines for osteoprogenitor cells and also promote their differentiation [14]. The US Food and Drug Administration has approved the use of recombinant human BMP-2 (rhBMP-2) in clinical practice. However, the low stability of the protein molecules requires the administration of supraphysiological doses to achieve a therapeutic effect. High doses and bolus type of growth factor release can lead to adverse reactions [15].
Reviewer 2 Report
Comments and Suggestions for Authors
Dear authors first I want to congratulate you for your research and second I want to make some suggestions in order to improve the scientific soundness of you paper.
1. Please provide some pictures with the structure of your scaffolds, especially because in the discussion part you specify, ''the fibrin clot in the scaffold structure imitates a hematoma'' (line 292)
2. Did you test the mechanical properties of the scaffolds? As far as I understand from your paper, you have developed scaffolds for guided bone regeneration, were the mechanical properties of the scaffolds are extremely important, because the scaffolds must widhstand the weight of the flap. You also stay in the discussion part '' Also, the addition of PRP to the collagen scaffold makes the hydrogel denser, allows it to maintain scaffold shape for several weeks" (line 325-327), how do you know the quantity of PRP is enough, without testing the mechanical properties?
3. Did you test the degradation of the scaffolds in vitro?
4. Put the "conclusion" title at the conclusion part
Author Response
Response to Reviewer
We would like to thank the reviewer for the useful comments and suggestions on our manuscript biomedicines-3242502, entitled “Collagen-PRP Mixed Hydrogels as a pBMP2 delivery system for bone defect regeneration”. We have checked and revised the manuscript carefully according to the comments and suggestions, and the detailed responses have been listed below with all the changes and improvements included in the revised manuscript.
Comments 1. Please provide some pictures with the structure of your scaffolds, especially because in the discussion part you specify, ''the fibrin clot in the scaffold structure imitates a hematoma'' (line 292).
Response 1: We added Figure 1, which shows the appearance of the material and its structure and explanations to it: “3.1. Col/PRP materials formation - Collagen hydrogel is unable to retain its shape. Adding PRP to Col made it possible to obtain a denser and resilient material (Figure 1a). Hematoxylin and eosin staining allowed to evaluate the structure of the materials (Figure 1b). We observed a uniform distribution of components in the structure of the matrices.”
Comments 2. Did you test the mechanical properties of the scaffolds? As far as I understand from your paper, you have developed scaffolds for guided bone regeneration, were the mechanical properties of the scaffolds are extremely important, because the scaffolds must widhstand the weight of the flap. You also stay in the discussion part '' Also, the addition of PRP to the collagen scaffold makes the hydrogel denser, allows it to maintain scaffold shape for several weeks" (line 325-327), how do you know the quantity of PRP is enough, without testing the mechanical properties?
Response 2: Yes, indeed, the physical and mechanical properties of materials are important when developing osteoplastic materials, but in this work we did not set the goal of obtaining a material similar to bone tissue. We developed a carrier matrix for delivering the vector system with BMP2 gene, which should support the new bone formation. Also, we added an image of the material (Figure 1) and showed that by mixing the components we got a more formed and elastic material.
Comments 3. Did you test the degradation of the scaffolds in vitro?
Response 3: In vitro degradation of the scaffolds was not studied in this work.
Comments 4. Put the "conclusion" title at the conclusion part
Response 4: We have added the "conclusion" at the conclusion part.
Reviewer 3 Report
Comments and Suggestions for Authors
The investigation of new biomaterials for bony defect regeneration is innovative and of significant scientific interest. The study is well-conceived, and the manuscript is generally well-written. However, prior to acceptance for publication, the following point needs to be addressed:
Please clarify the source of the PRP (Platelet-Rich Plasma) used in the study. Is it derived from rats, humans, or another source? This information is crucial for understanding the experimental design and its potential translational relevance.
Comments on the Quality of English LanguageN/A
Author Response
Response to Reviewer
We would like to thank the reviewer for the useful comments on our manuscript biomedicines-3242502, entitled “Collagen-PRP Mixed Hydrogels as a pBMP2 delivery system for bone defect regeneration”. The answer to the question is presented below. Explanations about the source of the PRP were also added to the manuscript.
Comments: Please clarify the source of the PRP (Platelet-Rich Plasma) used in the study. Is it derived from rats, humans, or another source? This information is crucial for understanding the experimental design and its potential translational relevance.
Response: The source of Platelet-Rich Plasma (PRP) in the work was rat blood, since the studies were carried out on a cell culture obtained from rat adipose tissue and implanted into rats. In this way, allogeneic materials were obtained that have lower immunogenicity compared to xenogeneic ones. When transitioning to clinical practice, PRP is planned to be derived from human blood to produce allogeneic or autologous materials. The source of PRP and its characteristics were clarified in the text of the article: “PRP was obtained from rat's blood using the previously described method [7]. The platelet content in PRP was 4.3±0.6 times higher than in whole blood”.
Reviewer 4 Report
Comments and Suggestions for Authors
The paper is a generally rigorous description of the production and biological analysis of Collagen-PRP Mixed Hydrogels used as pBMP2 delivery system and designed for use in bone defect regeneration. in vitro cytocompatibility study and an in vivo biocompatibility analysis are carried out with some promising results. The use of Col/PRP-based scaffolds was shown to be an effective method for delivering of osteoinductor gene to the site of bone tissue damage nd there was clear evidence that the highest degree of healing was observed after implantation of Col/PRP-TF/pBMP2 into the critical size defect.
Whilst there is sound biological characterisation, there is absolutely no materials characterisation. It is claimed within the bulk of the text that a scaffold structure has been produced, however I would argue that a scaffold structure should have some degree of porosity, should provide structural support for cell attachment and subsequent tissue development. This hydrogel may mimic some of the chemistry and biochemistry of the osteogenic microenvironment, but it absolutely doesn’t mimic the structure. There isn’t any particular issue in this and the title does not claim structural mimicry or a scaffold but I do believe that all mention of scaffold should be removed. The Combination of cytotoxicity analysis in vitro using well insert and in vivo biocompatibility is sensible approach to take although would be beneficial to see some discussion around the applicability of the well insert approach. It would also be beneficial to acknowledge the concerns around BMP-2 and whether the genetic construct approach may alleviate this. It could also benefit from a greater discussion of what a genetic construct is.
In terms of specific points:
Abstract
Make sure every abbreviation is defined here – this paper will be of interest to a broad spectrum of people who may not be aware what ADSCs and TF stand for
Introduction
Line 45, 46 – collagen is a hydrogel, not used in the development of hydrogel materials and there is significant control in cross-linking to vary rate of degradation significantly (see Enea at al where degradation is too slow).
Line 50 – how does PRP increase mechanical strength?
Line – 58-58 – reference 13 doesn’t really state that BMP-2 is the most commonly investigated or the most effective.
Line 60 – rhBMP2 – recombinant human (this should be included)
Method
Insufficient detail is provided on the collagen – what form is the collagen when it is mixed with the PRP (soluble / insoluble, liquid/ solid what % solid loading). How much thrombin solution was added for polymerisation, was it consistent in each batch or was it the amount associated with some phase transition? If the values are all as stated for the polyplex then this should be clearly stated.
What is the source of the plasmid DNA?
Results
Figure 1 – these images need to be much larger in order to identify any features of interest. Hard to see anything within the dapi figures.
Cell viability is greater than 100% - this is conceivable but should be discussed
What release rate of plasmid DNA is desired?
It would be beneficial to clearly label the regions of new bone formation within the CT scans.
Discussion
Referencing effect described in previous work on collagen-PRP construct – reference 1 actually describes the process mentioned as the standard repair mechanism this is not a consequence of a collagen-fibrin construct or the incorporation of genetic construct. Otherwise the discussion is strong and the conclusions valid
Round 2
Reviewer 2 Report
Comments and Suggestions for Authors
I have no further comments!